# Meta-Analysis of the Effects of Plyometric Training on Lower Limb Explosive Strength in Adolescent Athletes

**DOI:** 10.3390/ijerph20031849

**Published:** 2023-01-19

**Authors:** Lunxin Chen, Zhiyong Zhang, Zijing Huang, Qun Yang, Chong Gao, Hongshen Ji, Jian Sun, Duanying Li

**Affiliations:** 1Digitalized Performance Training Laboratory, Guangzhou Sport University, Guangzhou 510500, China; 2Sports Training Institute, Guangzhou Sport University, Guangzhou 510500, China

**Keywords:** plyometric training, explosive strength, adolescents, athletes, meta-analysis

## Abstract

Background: Plyometric training is an effective training method to improve explosive strength. However, the ability to perform plyometric training in the adolescent population is still controversial, with insufficient meta-analyses about plyometric training on lower limb explosive strength in adolescent athletes. Objective: To investigate the influence of plyometric training on the explosive strength of lower limbs in adolescent athletes. Methods: We performed a search of six databases (Web of Science, PubMed, Scopus, ProQuest databases, China National Knowledge Infrastructure (CNKI), and Wan-fang database) from the starting year of inclusion in each database to April 4, 2022. The quality of the included literature was assessed using the Cochrane risk assessment tool, and data were analyzed using the Review Manager 5.4 software. Result: Plyometric training had significant effects on the performance of adolescent athletes in countermovement jump (MD = 2.74, 95% CI: 1.62, 3.85, *p* < 0.01), squat jump (MD = 4.37, 95% CI: 2.85, 5.90, *p* < 0.01), standing long jump (MD = 6.50, 95% CI: 4.62, 8.38, *p* < 0.01), 10-m sprint (MD = −0.04, 95% CI: −0.08, −0.00, *p* = 0.03), and 20-m sprint (MD = −0.12, 95% CI: −0.20, −0.04, *p* = 0.03); all had positive and statistically significant effects (*p* < 0.05). Conclusion: Plyometric training can significantly enhance the explosive strength of lower limbs in adolescent athletes.

## 1. Introduction

Plyometric training is a fast, explosive exercise that includes a pre-lengthening or reversing movement and a complete stretch-shortening cycle (SSC) [1]. The rational use of plyometric training can effectively improve muscle strength and explosive strength [2,3], and the generation of the explosive strength can be well-explained by mechanical and neurophysiological models [1]. In the mechanical model, the rapid elongation of the tendon increases its elastic potential energy, which is then stored [4,5,6], and when followed by a rapid centripetal contraction, the previously stored elastic potential energy is rapidly released, thus increasing the force output [4,5,6]. The neurological model involves the detrusor reflex to enhance the force of the centripetal contraction [7,8,9].

Plyometric training of the lower extremity can be used in almost all sports [10]. Whenever we talk about jumping or sprinting ability training, we associate it with plyometric training [11,12]. There are various training methods for plyometric training; in terms of training body parts, it can be divided into upper limbs, lower limbs, and trunk [10]. In terms of training movements, it can be divided into single-legged jump, double-legged jump, drop jump, etc. [10]. On the practice plane, it can be divided into hard surface, sand, grass, water, etc. [13,14,15,16]. Plyometric training is also widely used for its effectiveness; many studies have shown that plyometric training can effectively improve muscle strength, power, jumping performance, sprinting performance, etc. [17,18,19]. Although previous meta-analyses have shown that plyometric training can improve athletic performance including jumping and sprinting performance and lower body muscle strength [20,21,22], there is still much controversy when plyometric training is applied to a specific group of adolescents. Some studies have shown that the epiphyseal plates of prepubertal adolescents have not yet closed [23,24], so high-intensity lower extremity training such as deep jumping is inappropriate [25,26]. However, some studies have found that children as early as 7–8 years of age can progressively perform plyometric training and continue training into adolescence and adulthood [27].

The ability to perform plyometric training in the adolescent population is still controversial, with insufficient meta-analyses about plyometric training on lower limb explosive strength in adolescent athletes. Therefore, the purpose of this study was to investigate the effects of plyometric training on lower limb explosive strength in adolescent athletes (aged 10–19 years). Outcome indicators referred to countermovement jump (CMJ), squat jump (SJ), standing long jump (SLJ), 10-m sprint, and 20-m sprint. We hypothesized that plyometric training would be effective in improving lower limb explosive strength in adolescent athletes and would improve CMJ, SJ, DJ, 10-m sprint, and 20-m sprint.

## 2. Information and Research Methods

### 2.1. Search Strategy

The search databases included PubMed, Web of Science, Scopus, ProQuest, China National Knowledge Infrastructure (CNKI), and Wan-fang database, and the search period was from the starting year of inclusion in each database to 4 April 2022. English search keywords included plyometric, PT, Adolescent, Adolescence, Teens, Athletes, Professional Athletes, Elite Athletes, College Athletes, player, sprint performance, Countermovement jump, etc. The search keywords were identified by cross-checking and supplemented by manual searches, tracing the references included in the literature when necessary. PubMed was used as an example (Figure 1).

### 2.2. Study Selection

The literature inclusion criteria for this meta-analysis was based on the Subjects, Interventions, Controls, Outcomes and Studies (PICOS) format of evidence-based medicine.

Inclusion criteria were as follows: (1) subjects were adolescent athletes, and the age range of adolescents was based on World Health Organization standards (10–19 years old) [28]; (2) the types of training in the experimental group were all plyometric training combined with specific training, with total training hours being the same in each group; (3) control group as a blank control or for specific training; (4) the outcome indicators were all measures of explosive strength of the lower limbs, including CMJ, SJ, SLJ, 10-m sprint, and 20-m sprint; and (5) study types were all randomized controlled trials (RCTs).

Exclusion criteria were as follows: (1) non-randomized experiments, self-controlled experiments, and randomized crossover experiments; (2) lack of experimental data; (3) conference abstracts or comments; and (4) unavailability of the full text of the literature.

A total of 19 studies were ultimately included in the paper for analysis, and the specific inclusion and exclusion process can be seen in Figure 2.

### 2.3. Data Extraction

All literature screened by the database was imported into EndnoteX9 software (Thomson ResearchSoft, Stanford, CT, USA) and duplicates were removed. The literature screening process was performed by two researchers alone (Z.Z. AND Z.H.); when the two researchers disagreed, it was resolved by questioning the third researcher (D.L.), with 95% agreement between the two researchers.

The extraction included: (1) basic information of the included studies, including study name, author’s information, journal’s information, and publication date; (2) characteristics of the study subjects, such as sample size per group and age and gender of study participants; (3) intervention duration and other specific details; (4) main reasons of the risk of bias assessment; and (5) post-test data of the study subjects (outcome mean and standard deviation of indicators, etc.).

### 2.4. Assessment of Risk of Bias

The Cochrane risk of bias assessment tool was used to assess the quality of included literature; “low risk of bias”, “unknown risk of bias”, and “high risk of bias” was used to judge each indicator. The quality of the literature was assessed by the number of “low risk of bias”, with grade A when there were four or more “low risk of bias”, grade B when there were two to three, and grade C when there was only one or no “low risk of bias”.

### 2.5. Statistical Analysis

Review Manager 5.4 software (RevMan, The Nordic Cochrane Centre, Copenhagen, Denmark) was used to analyze data. The outcome indicator metrics were identical, and the units were converted and unified, so the weighted mean difference (WMD), 95% confidence interval, was selected for the combined effect size indicator for data analysis. I^2^ was used to test for heterogeneity in each study, and when I^2^ was at 0–25%, heterogeneity was considered negligible, at 25–75%, moderate heterogeneity was considered to exist in the study, and at 75–100%, large heterogeneity was indicated in the study [29]. A fixed-effects model was used to analyze each outcome indicator when I^2^ was at 0–25%. A random-effects model was used to analyze each outcome indicator when I^2^ was at 25–75%.

## 3. Results

### 3.1. Study Characteristics

Nineteen trials were included in the analysis according to the PRISMA reporting guidelines, for a total of 21 sets of trials (Table 1). A total of 536 participants were included, and all participants were in the age range of 10–19 years. The intervention method for the experimental group was plyometric training with specific training, and the control group only performed specific training. The duration of each plyometric training intervention ranged from 10–35 min (mostly 20–25 min) and the frequency of intervention was mostly twice a week. Intervention periods ranged from 6–16 weeks, with 7–8 weeks being the most common.

### 3.2. Risk of Bias in the Included Articles

The Cochrane risk assessment tool was used to evaluate the quality of included studies (Figure 3); all included studies were randomized controlled trials, three studies performed allocation concealment [30,32,33], three implemented blinding of participants [31,35,36], and all included studies were low risk.

### 3.3. Meta-Analysis Results

#### 3.3.1. Jumping Performance

The study included 15 studies from 14 publications to assess the CMJ performance of adolescent athletes after plyometric training (Figure 4), which contained 404 subjects. Meta-analysis showed that plyometric training had a moderate positive effect on the improvement of CMJ performance in adolescent athletes (MD = 2.74, 95% CI (1.62, 3.85)), with moderate heterogeneity (I^2^ = 40%) and statistical significance (*p* < 0.01), which indicates that plyometric training was effective in improving CMJ performance in adolescent athletes. 

The study included ten studies to assess the SJ performance of adolescent athletes after plyometric training (Figure 5), which contained 285 subjects. The results of the meta-analysis showed that plyometric training had a positive effect on the improvement of SJ performance in adolescent athletes (MD = 4.37, 95% CI (2.85, 5.90)), with moderate heterogeneity (I^2^ = 65%), and was statistically significant (*p* < 0.01), which indicates that plyometric training was effective in improving SJ performance in adolescent athletes.

The study included seven studies to assess the SLJ performance of adolescent athletes after plyometric training (Figure 6), which contained 134 subjects. Meta-analysis showed that plyometric training had a positive effect on the improvement of SLJ performance in adolescent athletes (MD = 6.50,95% CI (4.62, 8.38)), with no study heterogeneity (I^2^ = 0), and was statistically significant (*p* <0.01), which indicates that plyometric training was effective in improving SLJ performance in adolescent athletes.

#### 3.3.2. Speed Performance

The study included four studies to assess the 10-m sprint performance of adolescent athletes after plyometric training (Figure 7), which contained 128 subjects. Meta-analysis showed that there was a positive effect of plyometric training on the improvement of 10-m sprint performance in adolescent athletes (MD = −0.04,95% CI (−0.08, −0.00)). The study was not heterogeneous (I^2^ = 0%) and was statistically significant (*p* = 0.03), which indicates that plyometric training was effective in improving the ability of young athletes in the 10-m sprint.

The study included nine studies from eight publications to assess the 20-m sprint performance of adolescent athletes after plyometric training (Figure 8), which contained 247 subjects. Meta-analysis revealed a high positive effect of plyometric training on the improvement of 20m sprint performance in adolescent athletes (MD = −0.12, 95% CI (−0.20, −0.04)), with moderate study heterogeneity (I^2^ = 34%), and was statistically significant (*p* = 0.0007), which indicates that plyometric training was effective in improving the ability of adolescent athletes in the 20-m sprint.

### 3.4. Reporting Bias

The funnel plot shows that the studies are more symmetrical, indicating a low risk of publication bias (Figure 9).

## 4. Discussion

### 4.1. Jumping Performance

Meta-analysis showed that plyometric training is an effective method in improving the jumping performance of adolescent athletes. The results are similar to the meta-analyses by Villarreal [22], Oxfeldt [21], Stojanovic [20], and Zhang Hao [49], in which CMJ and SJ were used as the outcome indicators to evaluate the effects of plyometric training on jumping performance in adolescents, and the subjects included in the studies were all adults. The subjects included in this study were all adolescent athletes, and the outcome indicators were CMJ, SJ, and SLJ to enrich the field of plyometric training.

The majority of meta-analyses and studies support that plyometric training has a positive effect on jumping performance [20,21,22]. Potdevin [41] et al. conducted a six-week experiment on adolescent swimmers and the results showed that a six-week plyometric training intervention was effective in improving the CMJ and SJ performance of the athletes in the experimental group, which had exactly the same growth and physical evolution as the control group, so the above results can be considered as a result of a significant increase in the maximum strength of the legs. There are various reasons for the increase in maximum muscle strength due to plyometric training, including changes in muscle structure due to increases in fascicle angle and fascicle length [50,51] and changes in stiffness of various elastic components (i.e., plantar flexor tendon complex) [52].

Furthermore, other studies showed that the significant improvement in jump height was primarily caused by neural adaptations [41,42,53,54,55,56], which was demonstrated by enhanced activation of motor units in the lower limb muscles (i.e., intramuscular coordination) and improved intermuscular coordination with reduced synergistic activation of antagonistic muscles [57]. Other neuroadaptive reasons for the increase in jump height due to plyometric training may be changes in stretch reflex excitability [56].

Plyometric training has different effects on adolescents at different maturity status. Many studies have concluded that plyometric training is more effective for athletes in pre-PHV (peak height velocity) [44,58]. Compared with adolescents in pre-PHV, those at mid-PHV have stiffness higher than pre-PHV in terms of vertical tendon stiffness (33.3%), leg stiffness (22%), eccentric muscle action (89.6%), and concentric muscle action (56.6%) [59]. Furthermore, during the peak of adolescent growth, accelerated growth can affect coordination and balance as the body is in a rapid growth phase and the trunk and legs develop at different times [60,61]. In contrast, the emergence of growth-spurt-related adolescent awkwardness in certain individuals may have a negative impact on jump performance during the mid-PHV phase [58,62]. Physical performance may also be limited by a decline in relative strength associated with increased body size throughout the mid-PHV phase, which can persist despite gains in absolute strength [63].

Volume and frequency are important indicators to measure the effectiveness of training. A study [42] showed that there was no significant difference in the plyometric training group at week four, showing improvement at week eight and an increase in CMJ and SJ performance from 16.7% to 25.9% after 12 weeks. However, a meta-analysis revealed that plyometric training performed three times per week for ten weeks provided superior training benefits [22]. We used to believe that the more we practiced, the better we would get; however, a study [64] claimed that plyometric training with moderate training frequency and volume (2 days per week, 840 jumps per week) produced the same training effect as high frequency and volume (4 days per week, 1680 jumps per week). These findings also suggest that there is a threshold of maximal training volume beyond which increases in volume are no longer beneficial [22].

Notably, there was an experiment showing negative effects after the plyometric training intervention. In contrast, in the SJ, a group emerged that improved more significantly after the intervention [33,36]. These two studies are also the main source of heterogeneity. Meszler [36] et al. found that a seven-week, twice-weekly plyometric training did not improve jumping ability in adolescent female basketball players, but rather decreased it, possibly due to the overuse of plyometric training. Fischetti [33] et al. conducted an eight-week, twice-weekly plyometric training for adolescent track and field athletes. The results of the study found that the combination of plyometric training with specific training produced a highly significant improvement in the athletes’ jumping ability (SJ) compared with sprint-specific training, and the significant improvement may be due to the specificity of the plyometric training, which mainly involves jumping in the vertical direction rather than the horizontal direction.

In conclusion, plyometric training can effectively improve jump performance in adolescent athletes. In the case of adolescents, the improvement in jump performance is mainly caused by neural adaptation. For adolescent athletes of different ages, the improvement in jumping performance is more significant in early adolescents, so it is reasonable for preadolescence to participate in plyometric training.

### 4.2. Sprint Performance

The meta-analysis showed that plyometric training is an effective method in improving sprint performance in adolescent athletes. The finding that plyometric training were effective in improving sprint performance is likewise similar to the findings of Ramirez-Campillo [65] et al., Oxfeldt [21] et al., and Zhang Hao [49] et al.

The results of the meta-analysis, similar to most of the included literature, concluded that plyometric training had positive effects on sprint performance in adolescent athletes. Negra [38] et al. conducted a twice-weekly plyometric training intervention with adolescent soccer players for a total of 12 weeks, and found that sprint performance was significantly higher in the plyometric training group versus the resistance training group than in the control group after the intervention. The improvement may be due to the increase in muscle maximal strength or explosive strength after resistance training or plyometric training, which allows the athletes to explode with greater force right at the start and increases the stride length of the athletes [56,66,67].

Physiological analysis shows that plyometric training appears to have induced peripheral and central neural adaptations as well as neuromuscular factor enhancement, resulting in improved joint position sense and detection of joint motion. Plyometric training may have resulted in peripheral adaptations due to repetitive stimulation of articular mechanoreceptors near the end range of motion [68,69]. The above reasons are the main reasons for plyometric training to improve sprint performance.

Expanding on the total intervention period, Fischetti [33] et al. concluded that in order to improve jumping and speed performance in adolescent athletes, the minimum duration of plyometric training must be at least 6 weeks [17,70]. In terms of speed improvement, the trained population was more responsive to various training stimulus than the untrained [71].

Similarly, opposite effects were observed in sprint performance for two studies compared with the other studies included. Negra [39] et al. conducted an eight-week, twice-weekly plyometric training intervention for adolescent soccer players, and showed that the improvement in 20-m sprint performance in the plyometric training group was less than in the soccer-specific training group, which may be related to soccer-specific training, which often involves technical movements such as accelerated movement, and thus soccer training may contribute to speed development [37]. Similar results appeared to emerge in an experiment by Villarreal [45] et al. In a seven-week, twice-weekly plyometric training intervention, the experimental and control groups did not show significant differences, and the reason for the difference in training effect may be due to the different adaptive responses of individuals to speed training [72]. It is well-known that sprint performance is the product of stride speed and stride length and therefore can be influenced by many factors, and since both factors are influenced by anthropometric characteristics, it may be one of the main possible explanations for the lack of improvement in sprint performance [45].

In summary, plyometric training is an effective means to improve sprint performance in adolescent athletes. Similar to jumping performance improvement, it is associated with an increase in maximum strength and explosive strength. It is important to keep the total training period in mind, and the training period for plyometric training should be at least six weeks. Those with training experience tend to have more significant improvements than those without.

## 5. Limitations

The majority of the literature included in this study had an intervention period of 7–8 weeks, twice a week. Although the study cited other studies to elaborate on the optimal intervention period, the evidence was insufficient to make it conclusive. In addition, the subjects included in this study were all adolescent athletes with training experience who were able to adapt to plyometric training better and faster compared with inexperienced or general individuals, and therefore cannot represent all adolescent populations.

## 6. Conclusions

This meta-analysis summarizes the existing studies on the effects of plyometric training on lower limb explosive strength in adolescent athletes, including soccer players, track and field athletes, and basketball players, etc. Our results show that plyometric training induces increased muscle maximal strength and neural adaptation and is an effective method for improving lower limb explosive strength (CMJ, SJ, SLJ, 10-m sprint, and 20-m sprint performance) in adolescent athletes; however, when it is implemented in adolescent athletes, we recommend that it be done under the supervision of qualified or experienced personnel. In addition, we believe that more studies on the benefits of different maturity status should be conducted so that adolescents at different maturity status can benefit from plyometric training.

## Figures and Tables

**Figure 1 ijerph-20-01849-f001:**
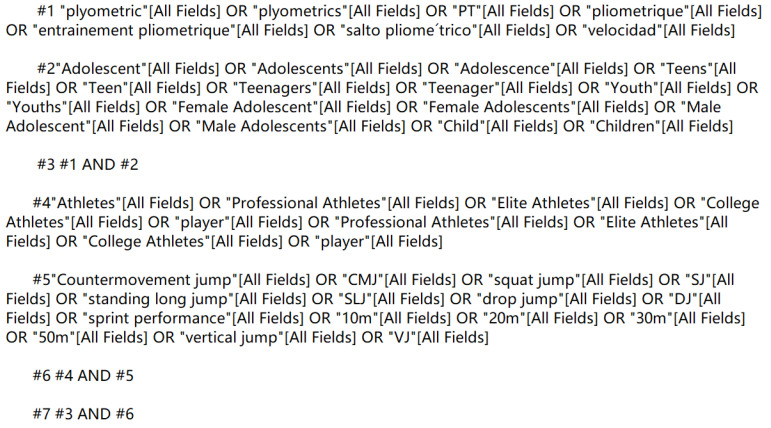
PubMed Literature Selection Strategy.

**Figure 2 ijerph-20-01849-f002:**
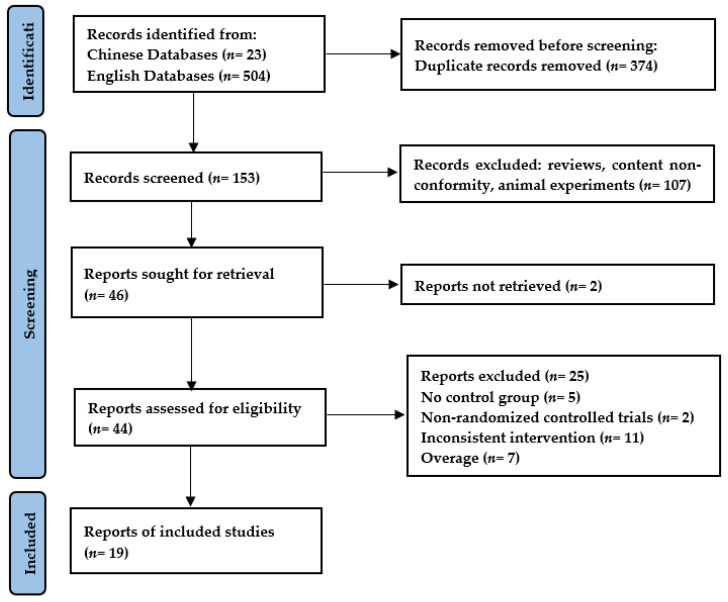
PRISMA flow chart for inclusion and exclusion of studies.

**Figure 3 ijerph-20-01849-f003:**
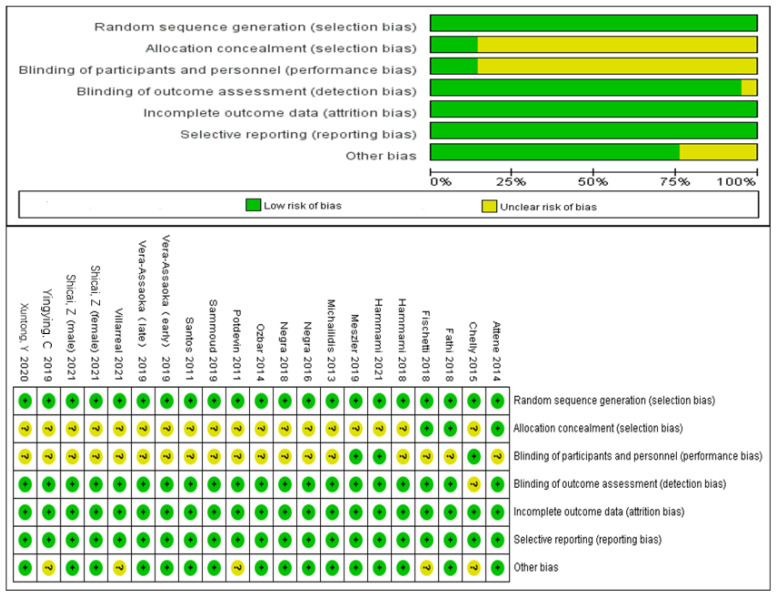
Judgments about each risk-of-bias item for each included study and risk-of-bias item presented as percentages across all included studies.

**Figure 4 ijerph-20-01849-f004:**
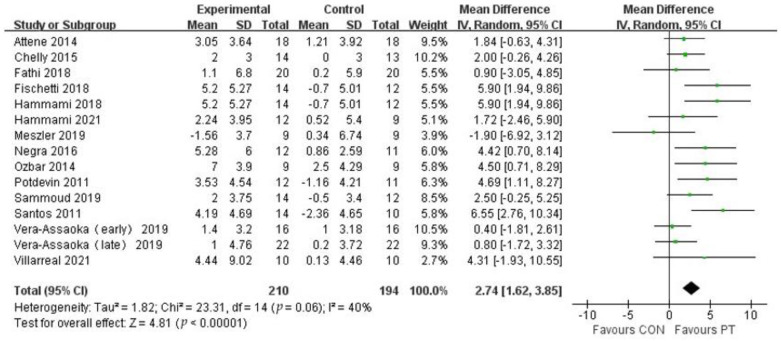
Forest plot of CMJ of experimental and control groups.

**Figure 5 ijerph-20-01849-f005:**
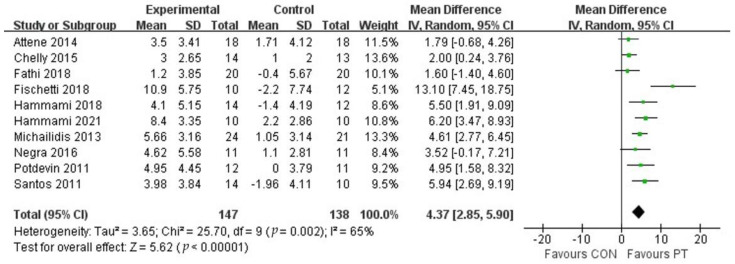
Forest plot of SJ of experimental and control groups.

**Figure 6 ijerph-20-01849-f006:**
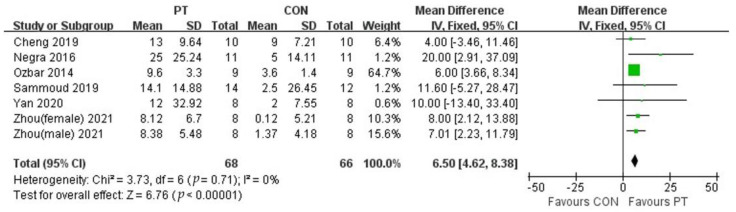
Forest plot of SLJ of experimental and control groups.

**Figure 7 ijerph-20-01849-f007:**
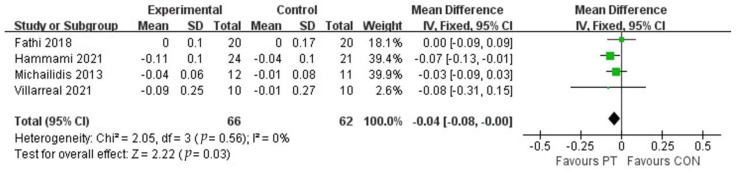
Forest plot of 10-m sprint of experimental and control groups.

**Figure 8 ijerph-20-01849-f008:**
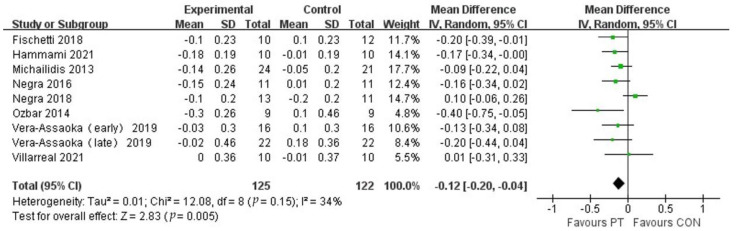
Forest plot of 20-m sprint of experimental and control groups.

**Figure 9 ijerph-20-01849-f009:**
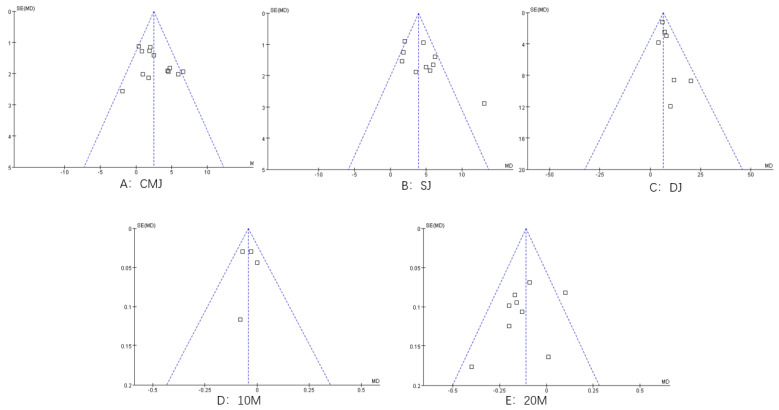
Plot of publication bias.

**Table 1 ijerph-20-01849-t001:** Characteristics of study participants.

Studies	Genders	Sample Size		Age		Experimental Group				Control Group	Key Outcome Indicators
		PT	CON	PT	CON	Interventions	PT Duration	Frequency	Duration	Interventions	
Attene [30] 2014	Female	18	18	14.9 ± 0.9	14.4 ± 0.7	Basketball + PT	20 min	2/week	6 weeks	Basketball	CMJ, SJ
Chelly [31] 2015	Male	14	13	11.9 ± 1.0	11.9 ± 1.0	Sprint + PT	20 min	4/week	10 weeks	sprint	CMJ, SJ
Fathi [32] 2018	Male	20	20	14.6 ± 0.5	14.5 ± 0.6	Volleyball + PT	25 min	2/week	16 weeks	Volleyball	CMJ, SJ, 10 m
Fischetti [33] 2018	Male	10	12	13.7 ± 0.5	13.5 ± 0.5	Track and Field + PT	15 min	2/week	8 weeks	Track and Field	CMJ, SJ, 20 m
Hammami [34] 2018	Male	14	12	15.7 ± 0.2	15.8 ± 0.2	Football + PT	25–30 min	2/week	8 weeks	Football	CMJ, SJ
Hammami [35] 2021	Male	10	10	16.4 ± 0.5	16.5 ± 0.4	Handball + PT	25–30 min	2/week	7 weeks	Handball	CMJ, SJ, 10 m, 20 m
Meszler [36] 2019	Female	9	9	15.8 ± 1.2	15.7 ± 1.3	Basketball + PT	20 min	2/week	7 weeks	Basketball	CMJ
Michailidis [37] 2013	Male	24	22	10.7 ± 0.7	10.6 ± 0.5	Football + PT	20–25 min	2/week	12 weeks	Football	SJ, 10 m, 20 m
Negra [38] 2016	Male	11	11	12.7 ± 0.3	12.8 ± 0.3	Football + PT	35–40 min	2/week	12 weeks	Football	CMJ, SJ, SLJ, 20 m
Negra [39] 2018	Male	13	11	12.7 ± 0.2	12.7 ± 0.2	Football + PT	25–35 min	Bi-weekly	8 weeks	Football	20 m
Ozbar [40] 2014	Female	9	9	18.3 ± 2.6	18.0 ± 2.0	Football + PT	60 min	Bi-weekly	8 weeks	Football	CMJ, SLJ, 20 m
Potdevin [41] 2011	Male/Female	12	11	14.3 ± 0.2	14.1 ± 0.2	Swim + PT	10 min	2/week	6 weeks	Swim	CMJ, SJ
Sammoud [42] 2019	Male	14	12	10.3 ± 0.4	10.5 ± 0.4	Swim + PT	20–35 min	2/week	8 weeks	Swim	CMJ, SLJ
Santos [43] 2011	Male	14	10	15.0 ± 0.5	14.5 ± 0.4	Basketball + PT	20 min	2/week	10 weeks	Basketball	CMJ, SJ
Vera-Assaoka [44] (early) 2019	Male	16	16	11.2 ± 0.8	11.5 ± 0.9	Football + PT	21 min	2/week	7 weeks	Football	CMJ, 20 m
Vera-Assaoka [44] (late) 2019	Male	22	22	14.4 ± 1.0	14.5 ± 1.1	Football + PT	21 min	2/week	7 weeks	Football	CMJ, 20 m
Villarreal [45] 2021	Male	10	10	13.57 ± 1.39	14.66 ± 0.86	Basketball + PT	20 min	2/week	7 weeks	Basketball	CMJ, 10 m, 20 m
Zhou [46](M) 2021	Male	8	8	10.01 ± 1.06	10.75 ± 1.03	Badminton + PT	20–30 min	3/week	8 weeks	Badminton	SLJ
Zhou [46](W) 2021	Female	8	8	10.01 ± 1.06	10.75 ± 1.03	Badminton + PT	20–30 min	3/week	8 weeks	Badminton	SLJ
Cheng [47] 2019	Male	10	10	16.5 ± 0.5	16.60 ± 0.52	Sprint + PT	90 min	3/week	12 weeks	sprint	SLJ
Yan [48] 2020	Male/Female	8	8	14.6 ± 0.7	14.6 ± 0.7	Badminton + PT	60 min	2/week	12 weeks	Badminton	SLJ

Attention: PT: plyometric training, CMJ: countermovement jump, SJ: squat jump, SLJ: standing long jump, 10 m: 10-m sprint, 20 m: 20-m sprint.

## Data Availability

All data generated or analyzed during this study are included in this published article.

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
