# Peer review of "Meta-Analysis of the Effects of Plyometric Training on Lower Limb Explosive Strength in Adolescent Athletes"

_ijerph, 2023, doi:10.3390/ijerph20031849_

Round 1

Reviewer 1 Report

Dear editor,

            I appreciate the opportunity to evaluate the manuscript "Meta-analysis of the effects of plyometric training on lower limb explosive strength in adolescent athletes". The aim of the manuscript was: to investigate the effects of plyometric training on lower limb explosive strength in adolescent athletes (aged 10-19 years). After reading, indicate that:

1 - The introduction needs to be more robust and indicate the state of the art in the research area (plyometrics).

2 - It is not clear which knowledge gap will be filled. There is little theoretical basis to support the objective of the research.

3 - The authors must indicate hypotheses for the study.

4 - The results should be divided according to the maturation stage, bearing in mind that this was the authors' theoretical foundation.

5 - The discussion should be more in-depth on each of the items addressed. Most of the authors do not present the mechanisms that provide improvement with plyometric training.

6 - The conclusion needs to present theoretical implications and go beyond what was written.

As it stands, the manuscript is not important to the field. However, if the authors go deeper into the topic addressed, I believe that the work has the potential to positively impact the area. Given the above, I am in favor of mandatory corrections.

Sincerely

Author Response

Dear Editors and Reviewers:

Thank you for your letter and for the reviewers’ comments concerning our manuscript entitled “Meta-analysis of the effects of plyometric training on lower limb explosive strength in adolescent athletes(ID: IJERPH-2126639). On behalf of my co-authors, we thank you very much for giving us an opportunity to revise our manuscript. Those comments are all valuable and very helpful for revising and improving our paper, as well as the important guiding significance to our researches. We have studied comments carefully and have made correction which we hope meet with approval. The main corrections in the paper and the responds to the reviewer’s comments are as flowing:

Responds to the reviewer’s comments:

  1. Response to comment:

The introduction needs to be more robust and indicate the state of the art in the research area (plyometrics).

Response:

Thank you for your response. After obtaining your suggestions, we have enriched the introduction section with the use of plyometric training, such as training body parts (upper limbs, lower limbs and trunk), training movements (single-legged jump, double-legged jump, drop jump, etc.), different training planes (hard surface, sand, grass, water, etc.) and the training benefits that come with plyometric training (muscle strength, power, jumping performance, sprinting performance, etc.).

  1. Response to comment:

It is not clear which knowledge gap will be filled. There is little theoretical basis to support the objective of the research.

Response:

Thank you for your careful observation. After our discussion, we decided that "fill the gap of this field" might not be a good expression for our article, so we changed the expression to "enrich the field of plyometric training" to make it more proper.

  1. Response to comment:

The authors must indicate hypotheses for the study.

Response:

After getting your comment, we decided to write the research hypothesis in the introduction section as a way to enrich the hierarchy of the article. We hypothesized that plyometric training would be effective in improving lower limb explosive strength in adolescent athletes, and would improve CMJ, SJ, DJ, 10-meter sprint, 20-meter sprint.

  1. Response to comment:

The results should be divided according to the maturation stage, bearing in mind that this was the authors' theoretical foundation.

Response:

Thank you for your professional advice. We likewise noticed a similar problem when we did the initial writing of our paper. After we extracted the data as well as classified all the included literature, we found that CMJ, SJ, and 20-meter sprint had more data and therefore had the opportunity to analyze them in subgroups. However, when we analyzed further we found that there were four studies in the CMJ group at PRE-PHV (peak height velocity), ten studies at MID-PHV, and two studies at POST-PHV, in addition to the total sample size of only 38 individuals at POST-PHV. The same trend was also found in the subgroup of SJ, 20 m sprint run. Therefore, we believe that the specificity of our included studies (adolescent athletes) resulted in a small number of studies at both PRE-PHV and POST-PHV, as well as a small total sample size, and therefore, we cannot guarantee that our results would be convincing if the different maturations stages were analyzed in subgroups. We thank you very much for being able to give us a good suggestion. We will continue to follow up the studies at different maturation stages and add them to our future studies.

  1. Response to comment:

The discussion should be more in-depth on each of the items addressed. Most of the authors do not present the mechanisms that provide improvement with plyometric training.

Response:

Thank you for your response. After double-checking the article, we found that in the "4.1 Jumping performance" section of the article, we have less description of the mechanisms underlying the plyometric training to improve jumping performance, Therefore, we have added the above-mentioned omissions in the second and third paragraphs of "4.1 Jumping performance". At the end of the second paragraph, we add two possible outcomes that may lead to increased muscle strength in the lower limbs, including changes in muscle structure (increases in fascicle angle and fascicle length) and changes in stiffness of various elastic components (plantar flexor tendon complex). At the end of the third paragraph we also add the neuroadaptive reasons (changes in stretch reflex excitability) that lead to improved jump performance.

  1. Response to comment:

The conclusion needs to present theoretical implications and go beyond what was written.

Response:

Thank you for your response. We have made changes to the discussion section in response to your suggestions. We have added theoretical implications for the conclusion section, which is “Our results show that plyometric training induces increased muscle maximal strength and neural adaptation and is an effective method for improving lower limb explosive strength (CMJ, SJ, SLJ, 10-meter sprint, and 20-meter sprint performance) in youth athletes”. And we point out some of our recommendations for future research, we believe that more studies on the benefits of different maturity status should be conducted so that adolescents at different maturity status can benefit from plyometric training.

We tried our best to improve the manuscript and made some changes in the manuscript. These changes will not influence the content and framework of the paper.

We appreciate for Editors/Reviewers’ warm work earnestly, and hope that the correction will meet with approval.

Once again, thank you very much for your comments and suggestions.

Reviewer 2 Report

Minor formatting changes needed throughout. Tables not properly fit to page (example table 1).

224 formatting changes required

252 253 odd phrasing and capitalization

293-314 citations need to be superscripted

Author Response

Dear Editors and Reviewers:

Thank you for your letter and for the reviewers’ comments concerning our manuscript entitled “Meta-analysis of the effects of plyometric training on lower limb explosive strength in adolescent athletes(ID: IJERPH-2126639). On behalf of my co-authors, we thank you very much for giving us an opportunity to revise our manuscript. Those comments are all valuable and very helpful for revising and improving our paper, as well as the important guiding significance to our researches. We have studied comments carefully and have made correction which we hope meet with approval. The main corrections in the paper and the responds to the reviewer’s comments are as flowing:

Responds to the reviewer’s comments:

  1. Response to comment:

Tables not properly fit to page (example table 1).

Response:

Thanks for your careful checks. We have made changes to the tables in response to your comments and have checked other tables and figures throughout the text.

  1. Response to comment:

224 formatting changes required.

Response:

We have completed the formatting of 224 and checked and standardized the formatting of the full text.

  1. Response to comment:

252 253 odd phrasing and capitalization

Response:

Thank you for your careful observation. We have modified the strange sentences and capitalization.

  1. Response to comment:

293-314 citations need to be superscripted.

Response:

Thank you for your response. All references of 293-314 have been superscripted.

When we checked the manuscript we found that references 46-49 were Chinese references, so we translated them into English, In addition, we found that the first appearance of CMJ, SJ, SLJ, in our manuscript did not indicate the full name, so we added it in the last paragraph of the introduction section

We tried our best to improve the manuscript and made some changes in the manuscript. These changes will not influence the content and framework of the paper.

We appreciate for Editors/Reviewers’ warm work earnestly, and hope that the correction will meet with approval.

Once again, thank you very much for your comments and suggestions.

Round 2

Reviewer 1 Report

Dear Editor,

          after the changes, I suggest approving the manuscript.